# ADVERSARIAL FEATURE DESENSITIZATION

## ABSTRACT

Deep neural networks can now perform many tasks that were once thought to be only feasible for humans. While reaching this impressive performance under standard settings, such networks are known to be vulnerable to adversarial attacks – slight but carefully constructed perturbations of the inputs which drastically decrease the network performance. Here we propose a new way to improve the network robustness against adversarial attacks by focusing on robust representation learning based on the adversarial learning paradigm, called here *Adversarial Feature Desensitization (AFD)*. AFD desensitizes the representation via an adversarial game between the embedding network and an additional adversarial discriminator, which is trained to distinguish between the clean and perturbed inputs from their high-level representations. Our method substantially improves the state-of-the-art in robust classification on MNIST, CIFAR10, and CIFAR100 datasets. More importantly, we demonstrate that AFD has better generalization ability than previous methods, as the learned features maintain their robustness across a wide range of perturbations, including perturbations not seen during training. These results indicate that reducing feature sensitivity is a promising approach for ameliorating the problem of adversarial attacks in deep neural networks.

## 1 INTRODUCTION

Despite remarkable recent progress in deep learning that allowed neural networks to achieve a near human-level performance across a range of complex tasks (He et al., 2016; Mnih et al., 2015; Silver et al., 2017; Vinyals et al., 2019), a number of important open challenges remain. For example, deep networks are know to be highly vulnerability to *adversarial attacks* (Szegedy et al., 2013), i.e. small but precise perturbations of the inputs that result in high-confidence predictions which are critically divergent from human judgement.

Many prior works on adversarial robustness have tackled the robust classification problem by forcing the classifier to output the correct label for the perturbed inputs (Madry et al., 2017; Kannan et al., 2018; Zhang et al., 2019b). These approaches essentially push the representations of samples from different categories away from the decision boundary. For example, the Adversarial Training procedure (Madry et al., 2017), trains a network to minimize the classification loss on the distribution of perturbed input samples. Another recent approach (Zhang et al., 2019b) augments the regular classification loss with an auxiliary term that encourages the network to match the assigned labels to clean and perturbed inputs (Figure 1a). More recently, several other works have tried to improve the classification robustness by enhancing the smoothness of the classification loss (Wu et al., 2019; Qin et al., 2020), or the saliency of the Jacobian matrix (Chan et al., 2020b). These methods has been shown to further improve the robust performance compared to prior approaches that do not consider the gradient landscape of the network. However, despite all these efforts, most of these defenses remain vulnerable against other forms of attacks that were not used during training or even slightly stronger perturbations of the same kind (Schott et al., 2018; Sitawarin et al., 2020).

One reason for the above could be an insufficient focus on the robustness of *representations* learned by the model. It has been shown that many adversarial perturbations that are often small in magnitude lead to large deviations in the high-level features of deep neural networks (Liao et al., 2018; Yoon et al., 2019). In addition, previous work (Ilyas et al., 2019) demonstrated that adversarial patterns often rely on specific learned features which generalize even on large datasets such as ImageNet (Deng et al., 2009). However, these features are highly sensitive to input changes, yielding a potential vulnerability that can be exploited by adversarial attacks. While humans can also experience altered

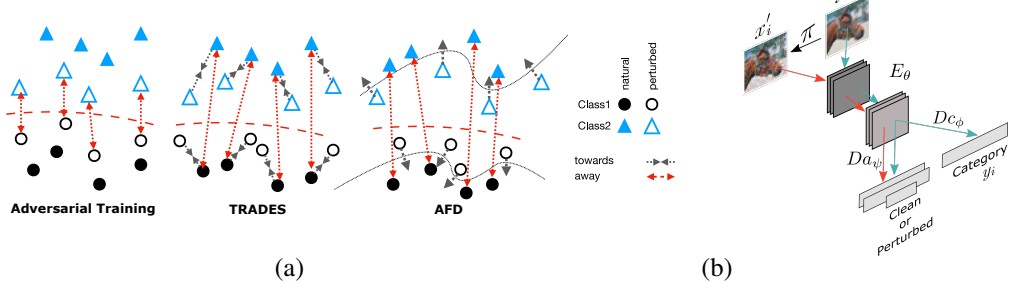

Figure 1: Overview of the proposed AFD approach: (a) visual comparison of several adversarial robustness methods (Adversarial training (Madry et al., 2017), TRADES (Zhang et al., 2019b), and AFD). The dotted black line corresponds to the decision boundary of the adversarial discriminator; (b) schematic of the proposed AFD paradigm.

perception in response to particular visual patterns (e.g., *visual illusions*[1]), they are seemingly insensitive to this particular class of perturbations, and often unaware of the subtle image changes resulting from adversarial attacks. This in turn suggests that current deep neural networks may rely on features that are still considerably different from those giving rise to perception in primates (and, particularly, in humans) – even despite many recent studies highlighting their remarkable similarities (Yamins et al., 2014; Khaligh-Razavi & Kriegeskorte, 2014; Bashivan et al., 2019). It is therefore reasonable to hypothesize that a deep network may become more robust to such adversarial attacks if the corresponding higher-level representations are more robust to input perturbations, similar to those used by our brains. One way to approach the issue of robust classification is to consider the classifier as a relatively simple mapping (e.g. a linear transformation) that produces predictions based on a learned representation. In this case, if the learned representation is robust then the predictions from the simple classifier would consequently be robust too (Garg et al., 2018; Zhu et al., 2020).

Here, instead of focusing on robust classification, we turned our attention to robustness of learned features from which the categories are inferred (e.g. using a simple linear classifier). Our goal is to learn representations that remain stable in the presence of adversarial attacks. *We propose to learn robust representations via an adversarial game between two agents: i) an attacker that searches for performance-degrading perturbations given the embedding function and ii) a discriminator function that distinguishes between the clean and perturbed inputs from their high-level representations.* The parameters of the embedding and the adversarial discriminator functions are then tuned via an adversarial game between the two (Figure 1b). This setup is similar to the adversarial learning paradigm widely used in image generation and transformation (Goodfellow et al., 2014a; Karras et al., 2019; Zhu et al., 2017), unsupervised and semi-supervised learning (Miyato et al., 2018b), video prediction (Mathieu et al., 2015; Lee et al., 2018), domain adaptation (Ganin & Lempitsky, 2015; Tzeng et al., 2017), active learning (Sinha et al., 2019), and continual learning (Ebrahimi et al., 2020). While some prior work have also considered adversarial learning to tackle the problem of adversarial examples, they have often been used to learn the distribution of the adversarial images(Wang & Yu, 2019; Matyasko & Chau, 2018), or the input gradients(Chan et al., 2020b;a).

The main contributions of this work are:

- We propose a novel method to learn adversarially robust representations through an adversarial game between the embedding function and an adversarial discriminator that distinguishes between the natural and perturbed representations.

- We theoretically show that our proposed adversarial approach leads to a flat loss function in the vicinity of the training samples, thereby making the overall representation more stable against adversarial attacks.

- We perform extensive empirical evaluations against many prior art methods, on three datasets, eight types of attacks, with a wide range of attack strength, and show that our proposed approach performs similar or better (often, significantly better) than most previous defense methods under most tested circumstances.

---

[1] https://michaelbach.de/ot/

## 2 METHODS

Let $E_\theta(x) : \mathcal{X} \to \mathcal{H}$, where $\mathcal{X} \subseteq \mathbb{R}^{N_i}$, $\mathcal{H} \subseteq \mathbb{R}^{N_e}$, be an *embedding function* (e.g. a neural network with parameters $\theta$) of the input $x \in \mathcal{X}$ into representation $h \in \mathcal{H}$, and let $Dc_\phi : \mathcal{H} \to \mathcal{Y}$, where $\mathcal{Y} \subseteq \mathbb{R}^{N_c}$, be a linear *decoding function*, with parameters $\phi$ (e.g., the last linear layer of a neural network before applying softmax). The likelihood of each class $i$ from a set of $N_c$ classes, $C = \{1, ..., N_c\}$, given the input $x$, is computed as follows: $l_i(x) = softmax\big(Dc_\phi(E_\theta(x))\big)_i, i \in C$. Let $\pi(x, \epsilon)$ denote a perturbation function (an adversarial attack) which computes the perturbed input $x'$ within the $\epsilon$-neighborhood of input $x$:

$$\forall x \in \mathcal{X} : \pi(x, \epsilon) = x' \in \mathcal{B}(x, \epsilon); \ \mathcal{B}(x, \epsilon) = \{x' \in \mathcal{X} : \|x' - x\| < \epsilon\}, \tag{1}$$

such that $\underset{i \in C}{\mathrm{argmax}} \, l_i(x) \neq \underset{i \in C}{\mathrm{argmax}} \, l_i(x')$, i.e. the attack changes the class label of a sample $x$.

It has been shown that adversarial examples are attributed to the presence of *non-robust* features which are predictive of the categories but are not shared with the human perception (Ilyas et al., 2019). Naturally, reducing the sensitivity of the learned features could potentially enhance the network classification robustness against adversarial attacks. Given the perturbation vector $\delta \in \mathbb{R}^{N_i}, \|\delta\| \leq \epsilon$, we could simply define the *sensitivity* of a representation as an empirical average (over $n$ input samples) of the maximum norm change in the representation due to input perturbation (attack): $S_e = \frac{1}{n} \sum_x \frac{1}{\epsilon} \max_\delta \|E(x) - E(x + \delta)\|$, and formulate the robust representation learning problem as an optimization problem which aims at minimizing the representation sensitivity $S_e$. However, such an approach may negatively affect the empirical risk objective, i.e. the classification accuracy (as we will later see in the empirical section). Thus, we desire a more precise formulation which would be less disruptive to the classification objective of the network.

### 2.1 ADVERSARIAL FEATURE DESENSITIZATION

Instead of minimizing the empirical average of representation sensitivity across all samples in the dataset (as formulated in the previous section), we focus on minimizing the representation sensitivity at the level of distributions which we expect to be less disruptive to the classification objective. For this, we propose an adversarial learning procedure similar to Generative Adversarial Networks (GAN) (Goodfellow et al., 2014a), in which the generator network is replaced by an embedding network $E_\theta$ that learns to map the clean and perturbed inputs into representations that are indistinguishable from each other. Similar to the original GAN setup, a discriminator network $Da_\psi$ is trained to distinguish between representations of clean and perturbed inputs. The training procedure involves three loss functions that are optimized sequentially. First, parameters of the embedding function $E_\theta$ and decoder $Dc_\phi$ are tuned to minimize the classification softmax entropy loss (on clean inputs). Second, parameters $\psi$ of the adversarial discriminator $Da_\psi$ are tuned to minimize the cross-entropy loss associated with discriminating natural and perturbed inputs conditioned on the natural labels. Lastly, parameters of the embedding function $E_\theta$ are adversarially tuned to maximize the cross-entropy from the second step. Algorithm 1 summarizes the proposed approach (also, see Figure 1b). The adversarial training framework involves a two-player minimax game (Chrysos et al., 2019) between $E_\theta$ and $Da_\psi$, with value function $V(E_\theta, Da_\psi)$:

$$V(E_\theta, Da_\psi) = \mathbb{E}_{p(y)}\big[\mathbb{E}_{p(x|y)}[\mathcal{S}(-Da_\psi(E_\theta(x), y))]\big] + \mathbb{E}_{q(y)}\big[\mathbb{E}_{q(x|y)}[\mathcal{S}(Da_\psi(E_\theta(x), y))]\big], \tag{2}$$

where $p$ and $q$ correspond to natural and perturbed distributions, and $\mathcal{S}$ denotes the softplus function. Chrysos et al. (2019) proves that the global minimum of the adversarial training criterion $V(E_\theta, Da_\psi)$ is achieved if and only if $p = q$; in our setting, $p = P(E_\theta(x), y)$ and $q = P(E_\theta(x'), y)$, i.e. achieving the global minimum in eq. 2 would imply that the representations of natural and perturbed images conditioned on the class label would belong to the same probability distribution. In that case, a Bayes optimal classifier would achieve the same error rate on the perturbed inputs as it would on the natural inputs. We use this fact below to prove that, when $V(E_\theta, Da_\psi)$ is at its global minimum, the gradient of the likelihood function becomes equal to zero; i.e. the adversarial attack will fail to change the class likelihoods.

Let $\pi(x, \epsilon)$ be a policy which computes the perturbed input $x'$ within the $\epsilon$ neighborhood of input $x$: $\pi(x, \epsilon) = x - \frac{\partial l_t}{\partial x} = x' \in \mathcal{B}(x, \epsilon)$ where $t$ denotes the target (ground truth) class index; and $\mathcal{S}(Da_\psi)$ be a discriminator function $\mathcal{H} \to \{0, 1\}$ that distinguishes between natural and perturbed representations; where $\mathcal{S}$ is the softplus function. The following theorem clarifies this property of our approach, which was not previously taken care of, at least not explicitly, by alternative methods.

---

**Algorithm 1:** AFD training procedure

---

**Input:** Attack policy $\pi$, mini-batch $B$ of size $m$, encoding network $E_\theta$, adversarial discriminator
network $Da_\psi$, decoder network $Dc_\phi$, softplus function $\mathcal{S}$, and learning rates $\alpha$, $\beta$, and $\gamma$.
Read mini-batch $B = \{(x_1, y_1), ..., (x_m, y_m)\}$
**repeat**

$\quad x' \leftarrow \pi(x, \epsilon)$

$\quad \mathcal{L}_{EDc} = -\frac{1}{m} \sum_{i=1}^{m} log\Big( softmax(-Dc_\phi(E_\theta(x_i)))_{y_i}\Big)$

$\quad \mathcal{L}_{Da} = \frac{1}{m} \sum_{i=1}^{m} \Big[ \mathcal{S}(-Da_\psi(E_\theta(x_i), y_i)) + \mathcal{S}(Da_\psi(E_\theta(x_i'), y_i))\Big]$

$\quad \mathcal{L}_E = \frac{1}{m} \sum_{i=1}^{m} \mathcal{S}(-Da_\psi(E_\theta(x_i'), y_i))$

$\quad (\theta, \phi) \leftarrow (\theta, \phi) - \alpha\nabla_{\theta, \phi}\mathcal{L}_{EDc}$

$\quad \psi \leftarrow \psi - \beta\nabla_\psi\mathcal{L}_{Da}$

$\quad \theta \leftarrow \theta - \gamma\nabla_\theta\mathcal{L}_E$

**until** *training converged*;

---

**Theorem 1.** *If the adversarial optimization of embedding and discriminator functions, $E_\theta$ and $Da_\psi$, converges to the global minimum $(\theta^*, \psi^*)$ of the training objective in equation 2, then the gradient of the true class (t) likelihood with respect to the input $x$ is zero at any $x \in \mathcal{X}$, i.e. $\frac{\partial l_t}{\partial x} = 0$.*

See appendix (6.2) for proof.

While the assumption of convergence to global optimum is a strong assumption, in practice, it is possible to derive a bound on the classifier's robust error in terms of its error on clean inputs and a divergence measure between the clean and perturbed representations (see 6.4 in the appendix).

## 3 EXPERIMENTS

### 3.1 ADVERSARIAL ATTACKS

We used a range of adversarial attacks in our experiments, using existing implementations in the Foolbox (Rauber et al., 2017) and Advertorch (Ding et al., 2019) packages. We validated the models against different variations of the Projected Gradient Descent (PGD) (Madry et al., 2017) $(L_\infty, L_2, L_1)$, Fast Gradient Sign Method (FGSM) (Goodfellow et al., 2014b), Momentum Iterative Method (MIM) (Dong et al., 2018), Decoupled Direction and Norm (DDN) (Rony et al., 2019), Deepfool (Moosavi-Dezfooli et al., 2016), and C&W (Carlini & Wagner, 2017) attacks. For each attack, we swept the $\epsilon$ value across a wide range and validated different models on each. Specific settings used for each perturbation are listed in Table-A2.

### 3.2 ADVERSARIAL ROBUSTNESS

We validated our proposed approach on learning robust representations on the MNIST (LeCun et al., 1998), CIFAR10, and CIFAR100 (Krizhevsky et al., 2009) datasets. We used the PGD-$L_\infty$ attack to perturb the inputs during training. $\epsilon$ was set to 0.3 and 0.031 for MNIST and CIFAR datasets respectively. We used the activations before the last linear layer as the high-level representations ($\mathcal{H}$) of the network. In all experiments, the adversarial discriminator network ($Da_\psi$) consisted of three fully connected layers with Leaky ReLU nonlinearity followed by a projection discriminator layer that incorporated the labels into the adversarial discriminator through a dot product operation (Miyato & Koyama, 2018). We compared several variations of the adversarial discriminator architecture and evaluated its effect on robust classification on MNIST dataset (Table A6). Increasing the depth of the adversarial discriminator and adding the projection discriminator layer drastically improved the robust classification accuracy. We verified that the adversarial discriminator could successfully discriminate between the clean and perturbed embeddings initially and that this performance was reduced during training (Figure A5). The number of hidden units in all layers of $Da_\psi$ were equal (64 for MNIST and 512 for CIFAR). We used spectral normalization (Miyato et al., 2018a) on all layers of $Da_\psi$. Further details of training for each experiment are listed in Table-A1. We used three separate learning rates for tuning the embedding $E_\theta$, adversarial discriminator $Da_\psi$, and decoder $Dc_\phi$ parameters. To find the best learning rates, we randomly split the CIFAR10 train set into a train and validation sets (45000 and 5000 images in train and validation sets respecively). We then carried

Table 1: Comparison of robust accuracy against various attacks on different datasets. For all attacks we used $\epsilon = 0.3$ and $\frac{8}{255}$ for MNIST and CIFAR10/CIFAR100 datasets respectively. † indicates replicated results. NT: natural training; AT: adversarial training; AFD: adversarial feature desensitization; WB: white-box attack; BB: black-box attack where the adversarial examples were produced by running the attack on the NT ResNet18 model. Numbers reported with $\mu \pm \sigma$ denote mean and std values over three independent runs with different random initialization. * RST(Carmon et al., 2019) additionally uses 500K unlabeled images during training.

| Method | Dataset | Network | Clean | $\text{PGD}_{L_\infty}$ (WB) | FGSM (WB) | $\text{PGD}_{L_\infty}$ (BB) | FGSM (BB) |
|---|---|---|---|---|---|---|---|
| AT(Madry et al., 2017) | | LeNet | 98.8 | 93.2 | 95.6 | 96.0 | 96.8 |
| TRADES(Zhang et al., 2019b) | | LeNet | **99.48** | 96.07 | - | - | - |
| ATES(Sitawarin et al., 2020) | | LeNet | 99.11 | 94.04 | - | - | - |
| ABS(Schott et al., 2018) | MNIST | LeNet | 99.0 | 13 | 34 | - | - |
| Defense-GAN(Samangouei et al., 2018) | | ConvNet | 99.20 | - | - | - | 93.23 |
| NT† | | RN18 | 98.80±0.11 | 0.0±0.0 | 2.90±0.07 | 11.25±3.60 | 20.48±2.91 |
| AT(Madry et al., 2017)† | | RN18 | 99.13±0.11 | 95.16±0.12 | 97.33±0.18 | **98.41±0.09** | **98.17±0.18** |
| TRADES(Zhang et al., 2019b)† | | RN18 | 98.84±0.18 | 80.69±9.44 | 91.75±2.7 | 97.11±0.63 | 86.4±8.22 |
| AFD | | RN18 | 99.15±0.12 | **96.13±0.35** | **97.70±0.34** | 98.31±0.18 | 97.92±0.31 |
| AT(Madry et al., 2017) | | RN18 | 87.3 | 45.8 | 56.1 | 86.0 | **85.6** |
| TRADES(Zhang et al., 2019b) | | RN18 | 84.92 | 56.61 | - | - | - |
| ATES(Sitawarin et al., 2020) | | WRN-34-10 | 86.84 | 55.06 | - | - | - |
| RLFAT(Song et al., 2020) | | WRN-32-10 | 82.72 | 58.75 | - | - | - |
| RST+(Wu et al., 2019; Carmon et al., 2019)* | | WRN-34-10 | 89.82 | 64.86 | 69.60 | **88.77** | **87.61** |
| LLR(Qin et al., 2020) | CIFAR10 | WRN-28-8 | 86.83 | 52.99 | - | - | - |
| YOPO(Zhang et al., 2019a) | | RN18 | 83.99 | 44.72 | - | - | - |
| JARN(Chan et al., 2020b) | | WRN-34-10 | 84.8 | 46.7 | 65.7 | 59.3 | 70.3 |
| NT† | | RN18 | **95.40** | 0.12 | 47.79 | 12.00 | 54.65 |
| AT(Madry et al., 2017)† | | RN18 | 83.58 | 41.05 | 50.12 | 83.20 | 82.88 |
| TRADES(Zhang et al., 2019b)† | | RN18 | 82.22 | 52.30 | 58.16 | 80.36 | 79.69 |
| Feature-scattering(Zhang & Wang, 2019) | | WRN-28-10 | 90.00 | 70.5 | 78.4 | - | - |
| AFD | | RN18 | 89.38±2.71 | **77.72±10.78** | **85.34±0.03** | 86.33±2.21 | 84.70±1.40 |
| NT† | | RN18 | **76.12** | 0.01 | 9.67 | 1.55 | 15.43 |
| AT(Madry et al., 2017)† | | RN18 | 55.78 | 20.39 | 25.09 | 53.83 | 53.25 |
| TRADES(Zhang et al., 2019b)† | CIFAR100 | RN18 | 55.48 | 27.36 | 30.46 | 54.13 | 53.16 |
| RLFAT(Song et al., 2020) | | WRN-32-10 | 56.70 | 31.99 | - | - | - |
| Feature-scattering(Zhang & Wang, 2019) | | WRN-28-10 | 73.9 | **47.2** | **61.0** | - | - |
| AFD | | RN18 | 62.35±5.70 | 44.88±8.30 | 49.52±6.73 | **63.63±4.12** | **54.90±0.42** |

out a grid-search using the train-validation sets and picked the learning rates with highest validation performance. As baseline, we used a re-implementation of adversarial training (AT) method (Madry et al., 2017) and the official code for TRADES[2] (Zhang et al., 2019b) and denoted these results with † in the tables.

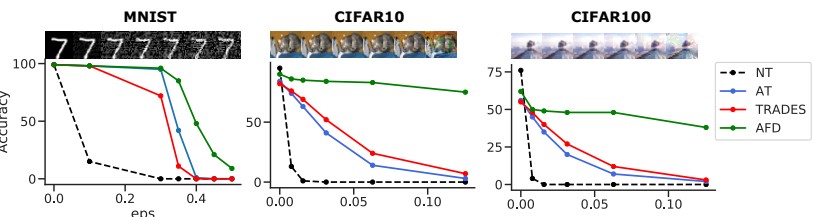

Figure 2: Robust accuracy for different strengths of PGD-$L_\infty$ attack on different datasets.

**Adversarial robustness against the observed attack** We first evaluated our approach against the same class and strength of attack that was used during training (PGD-$L_{\text{inf}}$ with $\epsilon = 0.3$ and 0.031 for MNIST and CIFAR datasets respectively). Table 1 compares the robust classification performance of our proposed approach against PGD-$L_\infty$ (with similar setting as was used during training) and FGSM attacks. Training LeNet with AFD was unstable leading to frequent crashing of adversarial discriminator accuracy despite our extensive hyperparameter search. For this reason, we conducted our MNIST experiments also using the ResNet18 architecture (He et al., 2016). On all datasets, AFD-trained network performed much better than alternative methods against both white-box and black-box attacks. The relative improvement was largest on CIFAR10 and CIFAR100 datasets. We also observed a relatively high variance in robust accuracy of AFD-trained networks on CIFAR datasets when trained from different random initializations (standard deviation of 10.78 and 8.30 for CIFAR10 and CIFAR100). We suspect this large variance to be due to the additional randomness in AFD training due to the adversarial game between the embedding and the adversarial discriminator networks. Across the three runs, the best trained models performed 83.72% and 54.95% against

---

[2]https://github.com/yaodongyu/TRADES.git

Table 2: AUC measures for different perturbations and methods on MNIST, CIFAR10, and CIFAR100 datasets. AUC values are normalized to have a maximum allowable value of 1. Evaluations on AT and TRADES were made on networks trained using reimplemented or official code.

| Dataset | Model | $PGD_{L_\infty}$ | $PGD_{L_2}$ | $PGD_{L_1}$ | FGSM | MIM | DDN | DeepFool | C&W |
|---|---|---|---|---|---|---|---|---|---|
| MNIST | NT | 0.16 | 0.12 | 0.14 | 0.29 | 0.16 | 0.18 | 0.23 | 0.75 |
| | AT | 0.67 | 0.50 | 0.44 | 0.76 | **0.84** | 0.72 | 0.74 | **0.96** |
| | TRADES | 0.62 | 0.47 | 0.41 | 0.72 | 0.80 | 0.77 | 0.73 | **0.96** |
| | AFD | **0.83** | **0.84** | **0.70** | **0.84** | **0.84** | **0.85** | **0.83** | **0.96** |
| CIFAR10 | NT | 0.04 | 0.02 | 0.06 | 0.35 | 0.04 | 0.06 | 0.08 | 0.13 |
| | AT | 0.27 | 0.05 | 0.14 | 0.39 | 0.28 | 0.06 | 0.33 | 0.41 |
| | TRADES | 0.34 | 0.06 | 0.16 | 0.46 | 0.36 | 0.06 | **0.41** | **0.47** |
| | AFD | **0.73** | **0.43** | **0.82** | **0.85** | **0.84** | **0.38** | 0.33 | 0.37 |
| CIFAR100 | NT | 0.03 | 0.02 | 0.03 | 0.09 | 0.02 | 0.05 | 0.03 | 0.10 |
| | AT | 0.15 | 0.03 | 0.08 | 0.19 | 0.15 | 0.04 | 0.16 | 0.23 |
| | TRADES | 0.19 | 0.04 | 0.09 | 0.23 | 0.20 | 0.04 | **0.19** | **0.26** |
| | AFD | **0.43** | **0.19** | **0.47** | **0.50** | **0.49** | **0.15** | 0.13 | 0.19 |

the white-box PGD-$L_{\text{inf}}$ attack ($\epsilon = 0.03$) on CIFAR10 and CIFAR100 respectively. Furthermore, AFD retained most of its robustness against a large set of attacks while improving robustness against C&W and DeepFool attacks when using particular weaker attacks (e.g. PGD-$L_\infty$ with $\epsilon = \frac{4}{255}$ and 5 iterations) during training (Figure-A6). In addition, we also evaluated the AFD model on transfer black-box attacks from AT and TRADES models which further showed higher robustness to those attacks too (Table-A4).

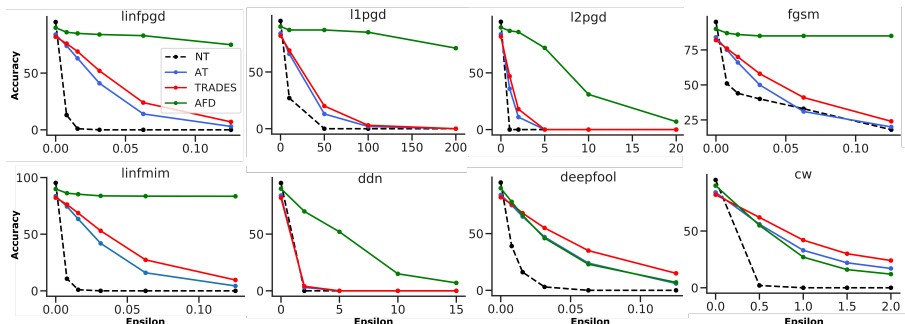

Figure 3: Comparison of robust accuracy of different methods against white-box attacks on CIFAR10 dataset with ResNet18 architecture.

**Robust classification against stronger and unseen attacks** We also validated the robustness classification against higher degrees of the same attack used during training as well as to a suite of other attacks that were not observed during training. We found that, compared to alternative defense methods, the AFD-trained networks continued to perform well against white-box attacks even for very large perturbations – while performance of other methods went down to zero relatively quickly (Figures 2,3,A1,A2). The AFD-trained network also performed remarkably well against most other attacks that were not observed during training (8/8 on MNIST and 6/8 on CIFAR datasets). To compare different models considering both attack types and perturbation strength, we computed the area-under-the-curve (AUC) for a range of epsilons for each attack and each approach. Table-2 summarizes these values for our approach and two alternative approaches (adversarial training and TRADES). Our results showed that compared to other baseline methods, AFD-trained networks are robust to a wide range of attacks and strengths. As discussed in the Methods section, unlike most previous defense methods that focus on minimizing the robust classification error, AFD minimizes the representation sensitivity and consequently, the learned representation remains stable for a large range of attack strengths compared to other methods (Figure 4-left).

Despite the large gain in robustness against most of the attacks, AFD-trained networks slightly underperformed against two of the attacks (Deepfool and C&W) when tested on CIFAR10 and CIFAR100 datasets. Our posthoc analyses showed that the direction of perturbations in the representational space in response to Deepfool and C&W attacks were more misaligned with the PGD-$L_{\text{inf}}$ attack compared to other attacks such as DDN which was comparatively less successful (Table A7). Moreover, it has been shown that most adversarial defenses are not guaranteed to transfer to unseen attacks (Maini et al., 2020; Pinot et al., 2020) and that different adversarial training methods might even overfit to

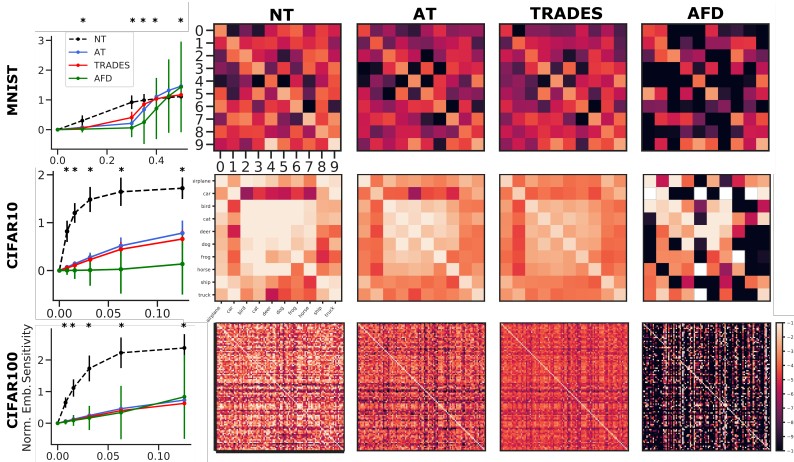

Figure 4: (left) Comparison of normalized representation sensitivity on test-set of MNIST (top), CIFAR10 (middle), CIFAR100 (bottom) datasets under PGD-$L_\infty$ attack. Plots show the median ($\pm$std) sensitivity over test-set for each dataset. * denotes statistically significant difference between sensitivity distributions for AFD and TRADES. (right) Logarithm of the average gradient magnitudes of class likelihoods with respect to input, evaluated at samples within the test-set of each dataset ($\log\left(\mathbb{E}_{x \sim \mathcal{X}}\left(\frac{\partial l_t}{\partial x}\right)\right)$). For each matrix, rows correspond to ground truth (target) labels and columns correspond to non-target labels.

the training set (Rice et al., 2020). While we did not observe any sign of overfitting for the PGD-$L_{\text{inf}}$ during training, the robustness against Deepfool and C&W attacks decreased during the later stages of training and in a way, the network might have overfit to the PGD-$L_{\text{inf}}$ attack during training (Figure A4).

**Representation sensitivity** We compared the robustness of the learned representation derived from training the same architecture using different methods. For that we measured the normalized sensitivity of the representations in each network as $\frac{\|E(x)-E(x')\|_2}{\|E(x)\|_2}$. For all three datasets we found that the AFD-trained networks learn high-level representations that were more robust against input perturbations as measured by the normalized L2 distance between clean and perturbed representations (Figures 4-left,A8,A9,A10).

**Gradient landscape** To empirically validate the prediction from Theorem-1, we computed the average gradient of class likelihoods with respect to the input across samples within the test set of each dataset ($\|\nabla_x l_i\|, i \in 1, ..., N_c$). We found that, on all datasets, the magnitude of gradients in the direction of most non-target classes were much smaller for AFD-trained network compared to other tested methods (Figure-4). This empirically confirms that AFD stabilizes the representation in a way that significantly reduces the gradients towards most non-target classes. Moreover, the output gradients of the AFD-trained network were highly salient and interpretable (Figure A7).

**Learning a sparse representation** As we discussed in the Methods section, we expected the AFD method to find and remove the non-robust features from the learned representation. Thus, we expected the learned representational space to potentially be of lower dimensionality. To test this, we compared the dimensionality of the learned representation using two measures. i) number of non-zero features over the test set within each dataset and ii) number of PCA dimensions that explains more than 99% of the variance in the representation computed over the test-set of each dataset. We found that the same network architecture when trained with AFD method gave rise to a much sparser and lower dimensional representational space (Table A5). The representational spaces learned with AFD on MNIST, CIFAR10, and CIFAR100 datasets had only 6, 9, and 76 principal components respectively.

**Adversarial vs. L2 optimization** We also ran an additional experiment on the MNIST dataset in which we added a regularization term to the classification loss to directly minimize the representation sensitivity $S_e = \frac{1}{n}\sum_x \|E(x) - E(x')\|$, during training. We observed that although this augmented loss led to learning robustness representations, it only achieved modest levels of robustness ($\sim 80\%$) and showed only weak generalization to stronger and other unseen attacks (Figure-A3). This result suggests that enforcing a distributional form of feature desensitization (e.g. AFD) may lead to robust behavior over a larger range of perturbations compared to the case where feature stability is directly enforced through an $L_p$ norm measure.

**Non-obfuscated gradients** Recent literature have pointed out that many defense methods against adversarial perturbations could drive the network into a regime called *obfuscated gradients* in which the network appears to be robust against common iterative adversarial attacks but could easily be broken using black-box or alternative attacks that do not rely on exact gradients (Papernot et al., 2017; Athalye et al., 2018; Carlini et al., 2019). We believe that our results are not due to obfuscated gradients for several reasons. i) For most perturbations, the model performance continues to decrease with increased epsilon (Figures-3,A1,A2); ii) The iterative perturbations were consistently more successful than single-step ones (Table-1); iii) Black-box attacks were significantly less successful than white-box attacks (Table-1); iv) The AFD-trained model performed similar or better than alternate methods against the Boundary attack (Brendel et al., 2018) – an attack which does not rely on the network gradients (Table-A3). In addition to these tests, we also evaluated the AFD performance on B&B (Brendel et al., 2018) and AutoAttack (Croce & Hein, 2020). On these attacks, AFD was consistently better than or equal to the baseline models on MNIST and CIFAR10 datasets but was less robust on the CIFAR100 dataset (Table-A3).

## 4 RELATED WORK

There is an extensive literature on mitigating susceptibility to adversarial perturbations. Adversarial training (Madry et al., 2017) is one of the earliest successful attempts to improve robustness of the learned representations to potential perturbations to the input pattern by solving a "saddle point" problem composed of an inner and outer adversarial optimization. A number of other works suggest additional losses instead of direct training on the perturbed inputs. TRADES (Zhang et al., 2019b) adds a regularization term to the cross-entropy loss which penalizes the network for assigning different labels to natural images and their corresponding perturbed images. (Qin et al., 2020) proposed an additional regularization term (local linearity regularizer) that encourages the classification loss to behave linearly around the training examples. (Wu et al., 2019) proposed to regularize the flatness of the loss to improve adversarial robustness.

Our work is closely related to the domain adaptation literature in which adversarial optimization has recently gained much attention (Ganin & Lempitsky, 2015; Liu et al., 2019; Tzeng et al., 2017). From this viewpoint one could consider the clean and perturbed inputs as two distinct domains for which a network aims to learn an invariant feature set. Although in our setting, i) the perturbed domain continuously evolves while the parameters of the embedding network are tuned; ii) unlike the usual setting in domain-adaptation problems, here we have access to the labels associated with samples from the perturbed (target) domain. Despite this, (Song et al., 2019) regularized the network to have similar logit values in response to clean and perturbed inputs and showed that this additional term leads to better robust generalization to unseen perturbations. Related to this, Adversarial Logit Pairing (Kannan et al., 2018) increases robustness by directly matching the logits for clean and adversarial inputs. Another line of work is on developing certified defenses which consist of methods with provable bounds over which the network is *certified* to operate robustly (Zhang et al., 2019c; Zhai et al., 2020; Cohen et al., 2019). While these approaches provide a sense of guarantee about the proposed defenses, they are usually prohibitively expensive to train, drastically reduce the performance of the network on natural images, and the empirical robustness gained against standard attacks are low.

## 5 DISCUSSION

We proposed a method to decrease the sensitivity of learned representations in neural networks using adversarial optimization. Decreasing the input-sensitivity of features has long been desired in training neural networks (Drucker & Le Cun, 1992) and has been suggested as a way to improve adversarial robustness (Ros & Doshi-Velez, 2018; Zhu et al., 2020). Our results show that AFD can be used to effectively reduce the input-sensitivity of network features with minimal interference with the classification objective and to improve robustness against a family of adversarial attacks. Successful feature desensitization was dependent on having a strong adversarial discriminator and maintaining a balance between the embedding and discriminator networks throughout training. With regards to the computational cost, while AFD requires three SGD updates per batch, the additional computational cost is not significantly higher than many prior methods when considering that most of the computational cost is associated with generating the adversarial examples during training.

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

## 6 APPENDIX

### 6.1 NETWORK ARCHITECTURES

For all experiments, we trained the ResNet18 architecture (He et al., 2016) using SGD optimizer with 0.9 momentum and learning rates as indicated in Table-A1, weight decay of $10^{-4}$, batch size of 128. All learning rates were reduced by a factor of 10 after scheduled epochs.

Table A1: Training hyperparameters for each dataset and network.

| Dataset | Model | $\mathbf{LR}_E$ | $\mathbf{LR}_{Da}$ | $\mathbf{LR}_{EDc}$ | weight decay | batch size | Num. Epochs | Scheduled Epochs |
|---|---|---|---|---|---|---|---|---|
| MNIST | | | | | | | 100 | [50, 80] |
| CIFAR-10 | ResNet18 | 0.5 | 0.1 | 0.1 | $10^{-4}$ | 128 | 900 | [400, 800] |
| CIFAR-100 | | | | | | | 900 | [400, 800] |

### 6.2 PROOF OF THEOREM 1

**Theorem 1.** *If the adversarial optimization of embedding and discriminator functions, $E_\theta$ and $Da_\psi$, converges to the global minimum $(\theta^*, \psi^*)$ of the training objective in equation 2, then the gradient of the true class (t) likelihood with respect to the input $x$ is zero at any $x \in \mathcal{X}$, i.e. $\frac{\partial l_t}{\partial x} = 0$.*

*Proof.* Assume $Dc_i, i \in C$ is a set of differentiable functions that implement the Bayes optimal classifier from the $E(x)$ representation (note that we will drop the subscripts in $E_\theta$ and $Da_\phi$ notation for simplicity), i.e.

$$\hat{y} = \operatorname{argmax}_i l_i \quad \text{and} \quad l_i = P(y_i|x) = softmax\big(Dc(E(x))\big)_i, y_i \in C. \tag{3}$$

Assuming that for the perturbed inputs $x' = \pi(x, \epsilon)$ the adversarial training of $E$ and $Da$ converges to its global minimum, from Proposition 2 of (Chrysos et al., 2019) we have:

$$\forall x \in \mathcal{X}, y \in \mathcal{Y} : P(E(x), y) = P(E(\pi(x, \epsilon)), y), \tag{4}$$

Following from Bayes rule we have:

$$P(y_i = t|E(x))P(E(x)) = P(y_i = t|E(x - \delta))P(E(x - \delta)), \ \delta = \frac{\partial l_t}{\partial x}, \tag{5}$$

From equation 4, the marginal distributions $P(E(x))$ and $P(E(x - \delta))$ should be equal which leads to:

$$P(y_i = t|E(x)) = P(y_i = t|E(x - \delta)), \tag{6}$$

which can only be true if $\frac{\partial l_t}{\partial x} = 0$. $\qquad\square$

### 6.3 ADVERSARIAL ATTACKS

We used a range of adversarial attacks in our experiments. Hyperparameters associated with each attack are listed in the table below. Implementation of these attacks were adopted from Foolbox (Rauber et al., 2017), AdverTorch (Ding et al., 2019) packages.

### 6.4 BOUND ON CLASSIFIER'S ROBUST ERROR

Considering the representation distributions in response to clean and perturbed inputs (of a particular class) as two distinct domains of inputs, it is straight forward to use the math from domain adaptation literature to derive a bound on the classifier's robust error (i.e. under the perturbed scenario). In this case, we can directly adapt *Theorem 2* in (Ben-David et al., 2010) to derive this bound.

If $\mathcal{D}_c$ and $\mathcal{D}_p$ are distributions of representations in response to clean and perturbed inputs of a particular class $y_i$ respectively. Let $\mathcal{U}_c$ and $\mathcal{U}_p$ be samples of size $m$ each, drawn from $\mathcal{D}_c$ and $\mathcal{D}_p$. Let $\mathcal{H}$ be a hypothesis space of VC dimension $d$, then for any $\delta \in (0, 1)$, with probability at least 1-$\delta$ (over the choice of the samples), for every $h \in \mathcal{H}$:

Table A2: Attack hyperparameters for each dataset and attack.

| Attack | Dataset | Steps | $\epsilon$ | More | Toolbox |
|--------|---------|-------|------------|------|---------|
| FGSM | MNIST
CIFAR | 1 | [0, 0.1, 0.3, 0.35, 0.4, 0.45, 0.5]
$[0, \frac{2}{255}, \frac{4}{255}, \frac{8}{255}, \frac{16}{255}, \frac{32}{255}, \frac{64}{255}]$ | -
- | Foolbox |
| PGD-$L_1$ | MNIST
CIFAR | 50 | [[0, 10, 50, 100, 200]] | step=0.025 | Foolbox |
| PGD-$L_2$ | MNIST
CIFAR | 50 | [0, 2, 5, 10] | step=0.025 | Foolbox |
| PGD-$L_\infty$ | MNIST
CIFAR | 40
20 | [0, 0.1, 0.3, 0.35, 0.4, 0.45, 0.5]
$[0, \frac{2}{255}, \frac{4}{255}, \frac{8}{255}, \frac{16}{255}, \frac{32}{255}]$ | step=0.033
step=$\frac{2}{255}$ | Foolbox |
| MIM | MNIST
CIFAR | 40 | [0, 0.1, 0.3, 0.5, 0.8, 1]
$[0, \frac{2}{255}, \frac{4}{255}, \frac{8}{255}, \frac{16}{255}, \frac{32}{255}]$ | -
- | AdverTorch |
| DDN | MNIST
CIFAR | 100 | [0, 1, 2, 5]
[0, 2, 5, 10, 15] | -
- | Foolbox |
| Deepfool | MNIST
CIFAR | 50 | [0, 0.1, 0.3, 0.35, 0.4, 0.45, 0.5]
$[0, \frac{2}{255}, \frac{4}{255}, \frac{8}{255}, \frac{16}{255}, \frac{32}{255}, \frac{64}{255}]$ | -
- | Foolbox |
| C&W | MNIST
CIFAR | 100 | [0, 0.5, 1, 1.5, 2] | stepsize=0.05 | Foolbox |

$$\xi_p(h) \leq \xi_c(h) + \frac{1}{2}\hat{d}_{\mathcal{H}\Delta\mathcal{H}}(\mathcal{U}_c, \mathcal{U}_p) + 4\sqrt{\frac{2d\log(2m) + \log(\frac{2}{\delta})}{m}} + \lambda$$

where $\xi_c$ and $\xi_p$ are the errors on clean and perturbed inputs, $\hat{d}_{\mathcal{H}\Delta\mathcal{H}}$ is the empirical $\mathcal{H}$-divergence (Ben-David et al., 2010), and $\lambda$ is the is the combined error of the ideal hypothesis $h^*$: $\lambda = \xi_c(h^*) + \xi_p(h^*)$.

Table A3: Comparison of robust accuracy against AutoAttack (Croce & Hein, 2020), Boundary attack (Brendel et al., 2018) with 5000 steps and $\epsilon = 2$, and B&B attack (Brendel et al., 2019). We tested the robust performance of each model on 100 random samples from each dataset's test-set.

| Dataset | Model | Method | AutoAttack | Boundary Attack | B&B |
|---------|-------|--------|------------|-----------------|-----|
| MNIST | RN18 | NT | 0 | 25 | 3 |
| | | AT | 88 | 63 | 92 |
| | | TRADES | 2 | 48 | 17 |
| | | AFD | 92 | 78 | 96 |
| CIFAR10 | RN18 | NT | 0.0 | 0 | 0 |
| | | AT | 34 | 51 | 36 |
| | | TRADES | 49 | 58 | 54 |
| | | AFD | 25 | 68 | 41 |
| CIFAR100 | RN18 | NT | 0 | 2 | 0 |
| | | AT | 24 | 32 | 27 |
| | | TRADES | 30 | 35 | 30 |
| | | AFD | 15 | 32 | 10 |

Table A4: Transfer black-box attack from ResNet18 network trained with adversarially training (AT) and TRADES on different datasets.

| Dataset | Method | AT Transfer | TRADES Transfer |
|---|---|---|---|
| MNIST | NT | 73.46 | 62.09 |
| | AT | 97.11 | 97.23 |
| | TRADES | 93.58 | - |
| | AFD | 97.48 | 97.63 |
| CIFAR10 | NT | 94.11 | 76.09 |
| | AT | 82.32 | 62.54 |
| | TRADES | 80.78 | - |
| | AFD | 88.56 | 65.13 |
| CIFAR100 | NT | 56.84 | 51.95 |
| | AT | - | 36.6 |
| | TRADES | 40.1 | - |
| | AFD | 42.72 | 40.29 |

Table A5: Dimensionality of the learned representation space on various datasets using different methods and measures. Units: number of non-zero feature dimensions over the test-set within each dataset. Dims: number of PCA dimensions that account for 99% of the variance across all images within the test-set of each dataset.

| Dataset | MNIST | | CIFAR10 | | CIFAR100 | |
|---|---|---|---|---|---|---|
| Network | RN18 | | RN18 | | RN18 | |
| | Units | Dims | Units | Dims | Units | Dims |
| NT | 64 | 24 | 512 | 97 | 512 | 429 |
| AT | 64 | 43 | 512 | 455 | 512 | 481 |
| TRADES | 64 | 40 | 512 | 349 | 512 | 461 |
| AFD | 18 | 6 | 417 | 9 | 500 | 76 |

Table A6: Comparison of robust accuracy against PGD-$L_\infty$ with $\epsilon = 0.3$ using different architectures for the adversarial discriminator, tested on MNIST dataset.

| Dataset | Model | $Da$ Architecture | Robust Acc. |
|---|---|---|---|
| MNIST | RN18 | FC1-PD | 85.96 |
| | | FC3 | 90.73 |
| | | FC3-PD | 97.03 |

Table A7: Comparison of representation perturbations in response to different attacks. We computed the cosine angle between representation perturbations due to each attack to those from PGD-$L_{inf}$. Values are reported in radians.

| Dataset | Model | Attack | Median Angle (rad) |
|---|---|---|---|
| MNIST | RN18 | DDN | 0.25 |
| | | C&W | 0.40 |
| | | Deepfool | 0.38 |
| CIFAR10 | RN18 | DDN | 1.03 |
| | | C&W | 1.13 |
| | | Deepfool | 1.12 |
| CIFAR100 | RN18 | DDN | 1.15 |
| | | C&W | 1.34 |
| | | Deepfool | 1.35 |

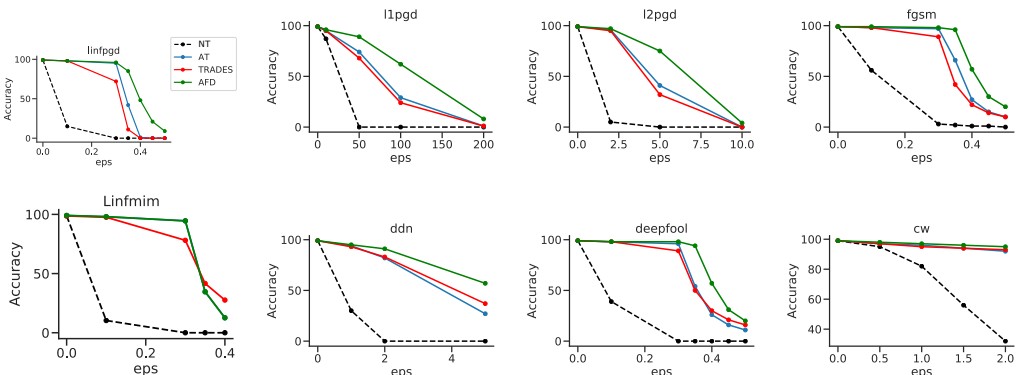

Figure A1: Comparison of robust accuracy of different methods against white-box attacks on MNIST dataset with ResNet18 architecture.

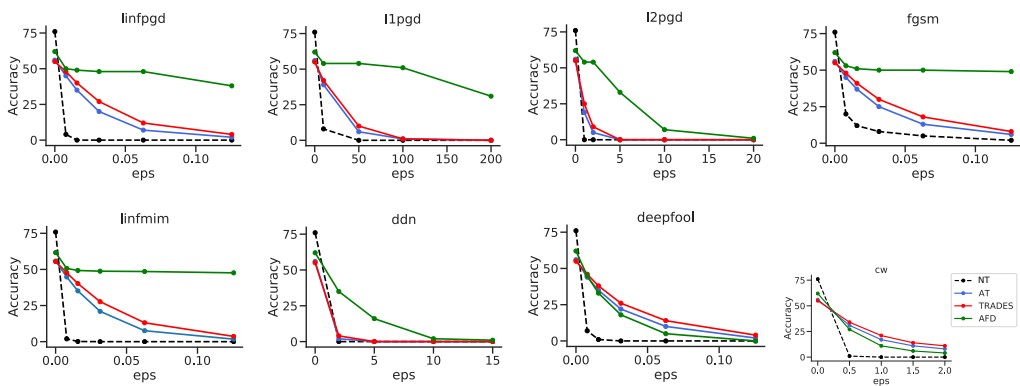

Figure A2: Comparison of robust accuracy of different methods against white-box attacks on CIFAR100 dataset with ResNet18 architecture.

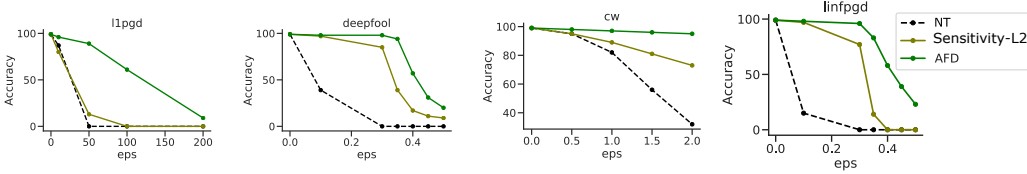

Figure A3: Comparison of robust accuracy of AFD and representation matching against white-box attacks on MNIST dataset with ResNet18 architecture.

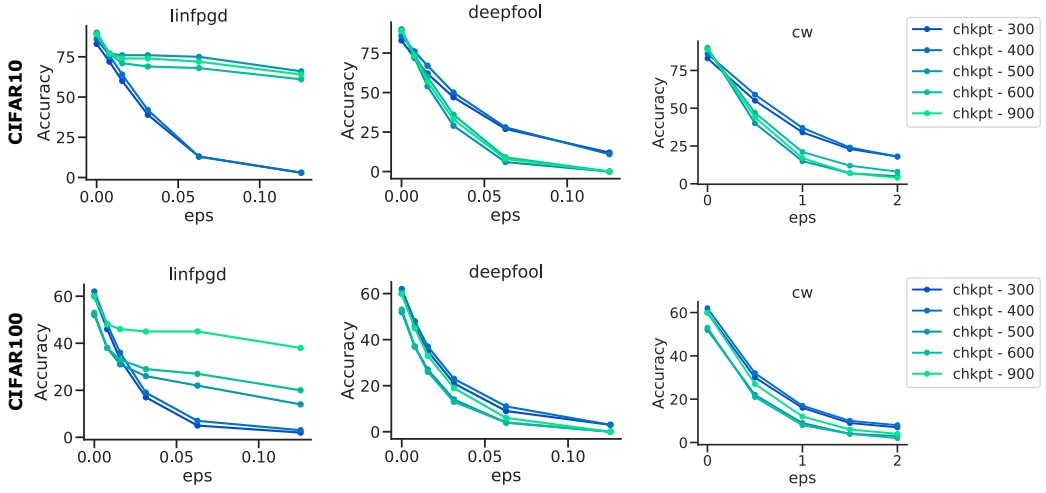

Figure A4: Robust accuracy against various attacks at different training stages. First and second rows correspond to models trained on CIFAR10 and CIFAR100 respectively.

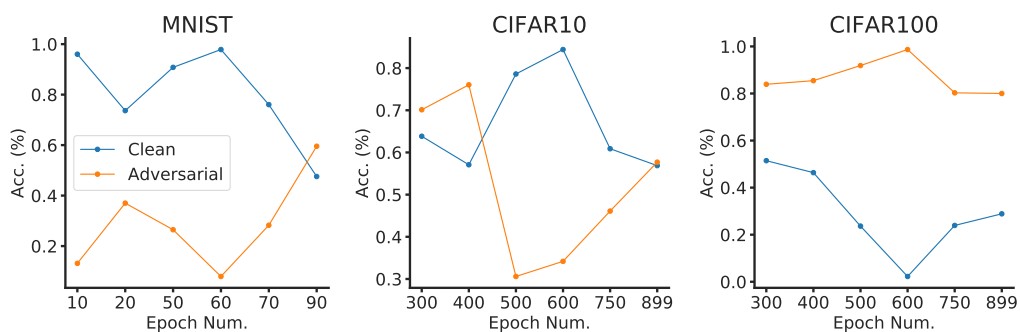

Figure A5: Classification accuracy of the adversarial discriminator $Da$ at different training stages on different datasets.

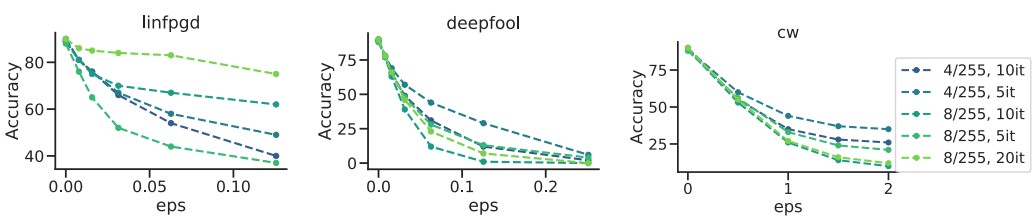

Figure A6: Robust accuracy of AFD-trained models on CIFAR10 dataset against various attacks when using different levels of attack strength during training.

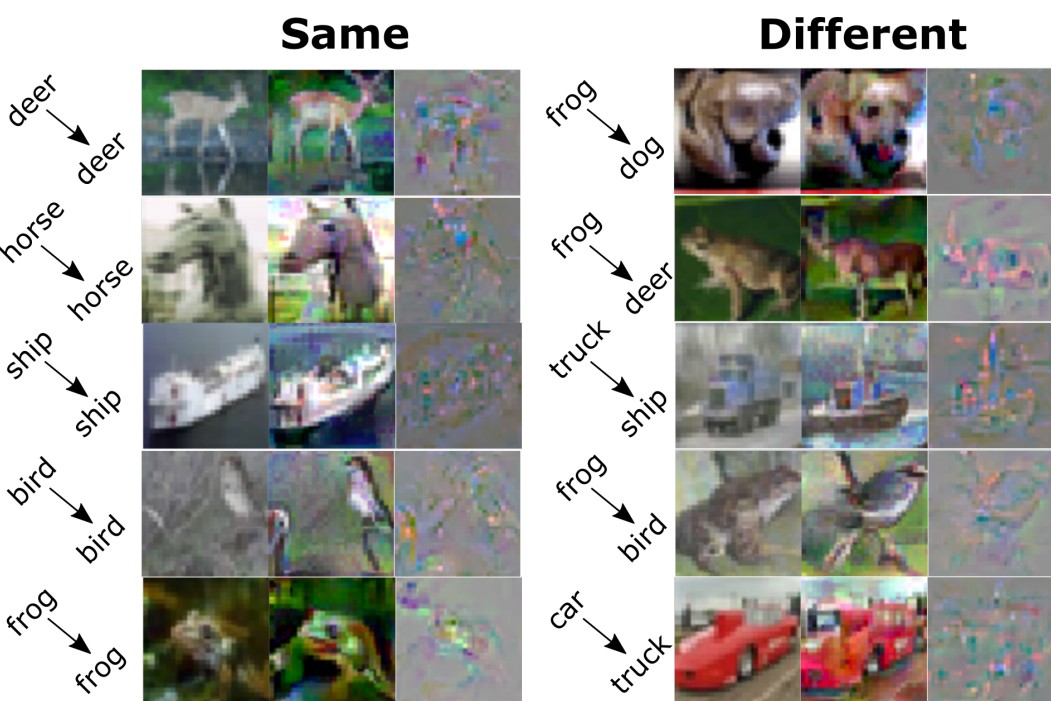

Figure A7: Feature visualization. Pixel values were changed in the direction of the gradients that would maximize either the ground truth class (left column) or a randomly selected class (right column). For each image, the original image (left), transformed image (middle) and gradient map (right) are shown.

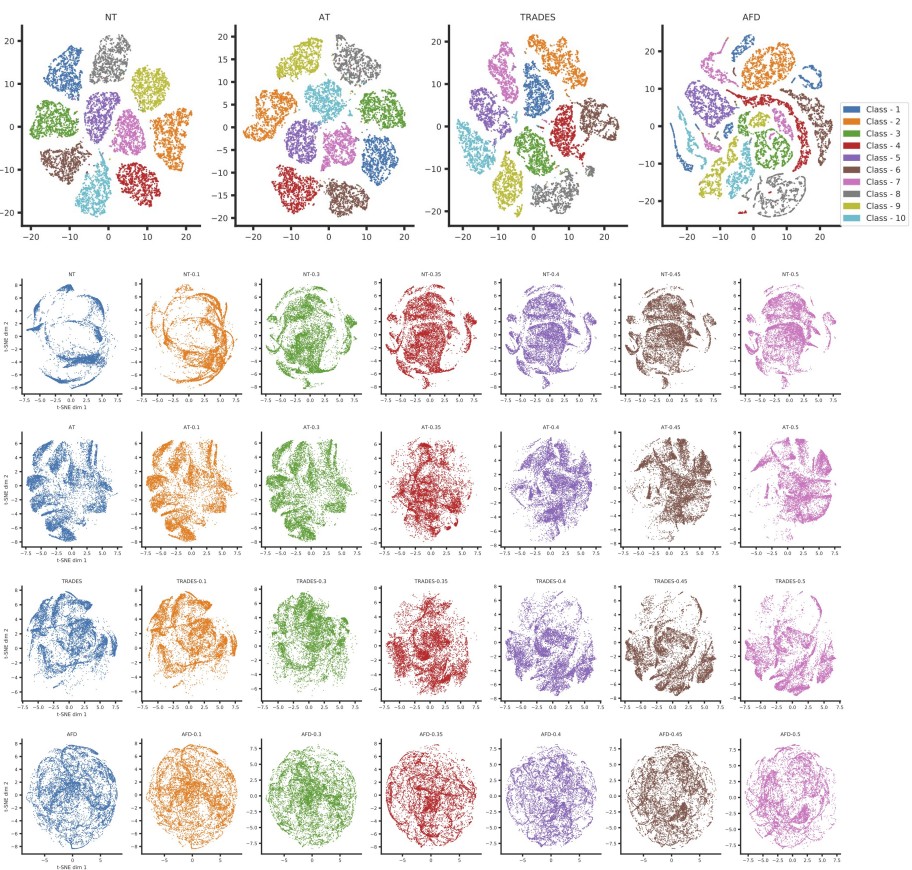

Figure A8: Scatter plot of 2-dimensional t-SNE projection (Maaten & Hinton, 2008) of the representations derived from training the ResNet18 architecture on MNIST dataset. (top row) t-SNE projection of representations of clean images for networks trained with different methods. Each point corresponds to the representation of one of the images from the MNIST test-set. (rows 2 to 5) t-SNE projection of the representation of the clean and perturbed MNIST test-set images. Columns are sorted from left to right with the strength of the perturbation (left-most column corresponds to clean images and right-most column with highest tested perturbation). Perturbations are generated using PGD-$L_\infty$ attack. NT: naturally trained; AT: adversarially trained(Madry et al., 2017); TRADES: (Zhang et al., 2019b); AFD: adversarial feature desensitization.

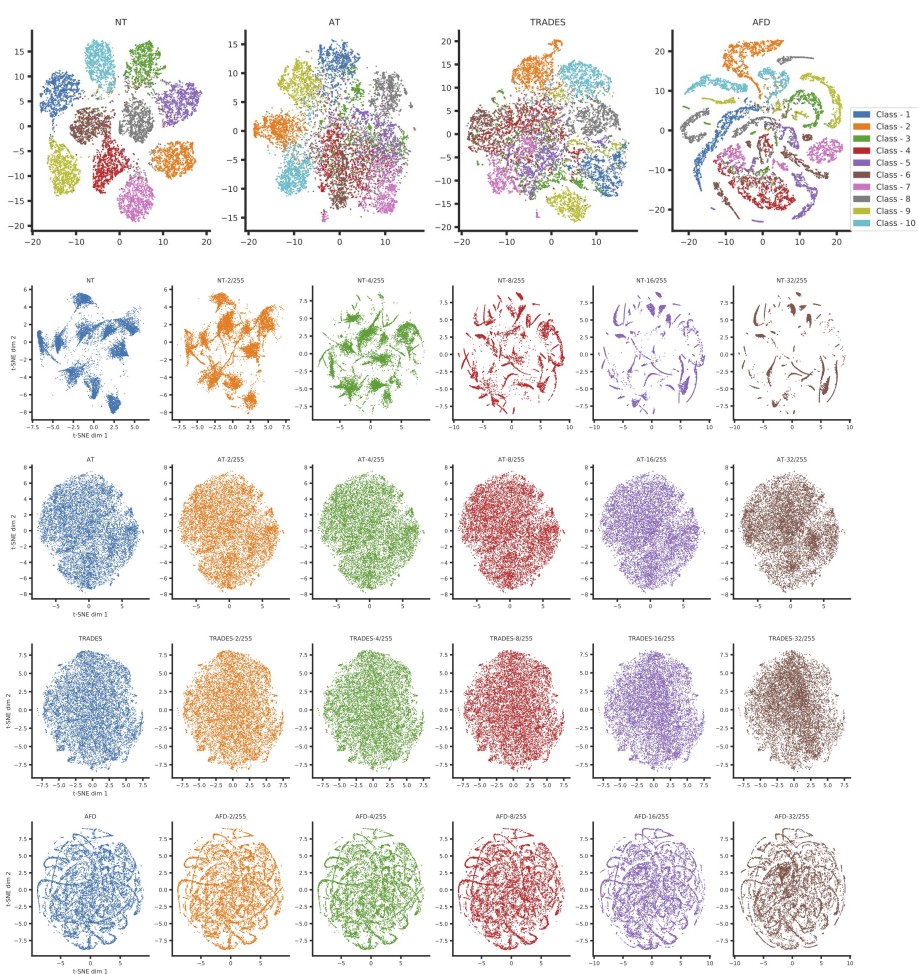

Figure A9: Scatter plot of 2-dimensional t-SNE projection (Maaten & Hinton, 2008) of the representations derived from training the ResNet5 architecture on CIFAR10 dataset. (top row) t-SNE projection of representations of clean images for networks trained with different methods. Each point corresponds to the embedding of one of the images from the CIFAR10 test-set. (rows 2 to 5) t-SNE projection of the embedding of the clean and perturbed CIFAR10 test-set images. Columns are sorted from left to right with the strength of the perturbation (left-most column corresponds to clean images and right-most column with highest tested perturbation). NT: naturally trained; AT: adversarially trained(Madry et al., 2017); TRADES: (Zhang et al., 2019b);AFD: adversarial feature desensitization.

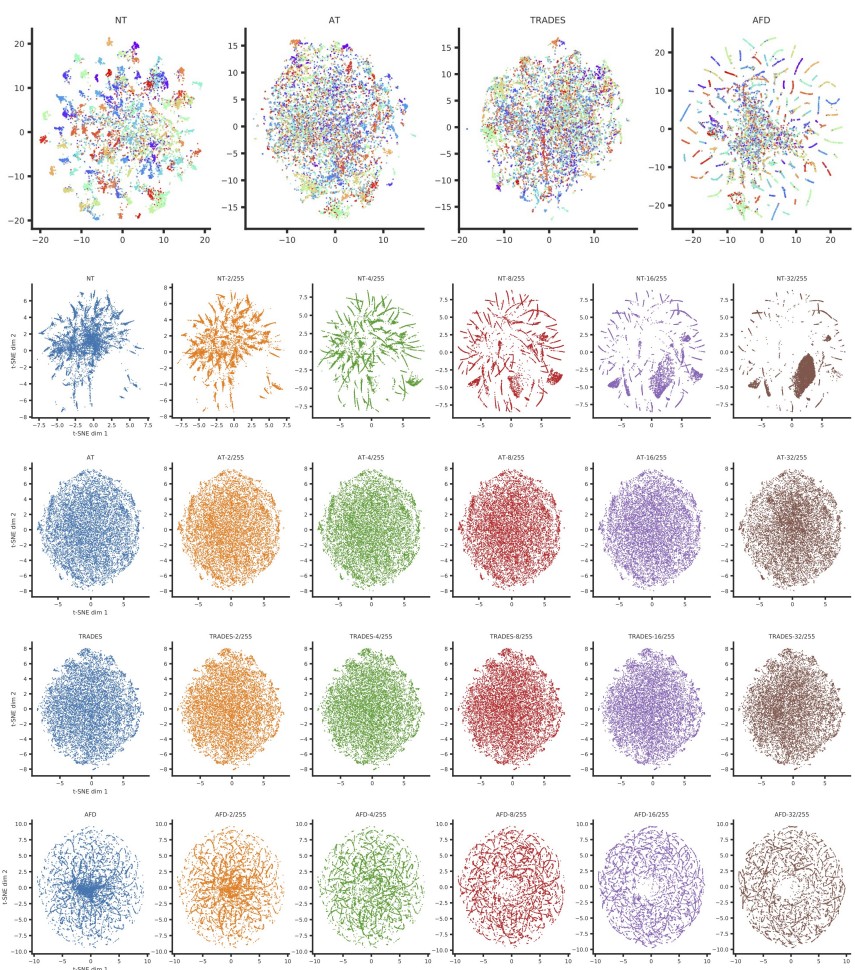

Figure A10: Scatter plot of 2-dimensional t-SNE projection (Maaten & Hinton, 2008) of the representation derived from training the ResNet5 architecture on CIFAR100 dataset. (top row) t-SNE projection of representations of clean images for networks trained with different methods. Each point corresponds to the representation of one of the images from the CIFAR100 test-set. (rows 2 to 5) t-SNE projection of the representation of the clean and perturbed CIFAR100 test-set images. Columns are sorted from left to right with the strength of the perturbation (left-most column corresponds to clean images and right-most column with highest tested perturbation). NT: naturally trained; AT: adversarially trained (Madry et al., 2017); TRADES (Zhang et al., 2019b); AFD: adversarial feature desensitization.

