# OpenReview forum: "Adversarial Feature Desensitization"
_ICLR.cc/2021/Conference — Reject_

### Official Review · AnonReviewer1 · 2020-10-14
**The main idea is reasonable. But the current manuscript may not be ready for publication.**

**Rating:** 4
**Confidence:** 4

**Review:**

The paper is about leveraging the GAN idea to get robust features that are insensitive to adversarial attacks. Specifically, with a shared encoder/embedding, the feature of the adversarial attack are considered as the fake sample, while the feature from the real image is the real sample; by forming an adversarial game in the feature space and by jointly considering the classification loss, the AFD method is proposed.

The main idea is reasonable. But the current manuscript may not be ready for publication.

The structure of the current manuscript should be revised substantially. For example, too many materials (like experimental results) are given in Appendix, which makes the main manuscript less self-contained; I would suggest rearranging the contents and put the important ones in the main manuscript.

Figures and their captions should be revised, especially Figures 3 and 4.


In the last paragraph of Page 2, the authors stated that “the proposed method outperforms existing methods by a large margin. However, this might not be true, by considering the performance shown in Tables 1 and 2 and the high variance observed in Table 1 and Figure 4.

In the 5th line of the first paragraph of Section 2, the definition of l_i (x) is different from the one in Algorithm 1.

In the last paragraph of Page 3, the definition of pi(x, epsilon) is not consistent. Epsilon is missing.

In the last paragraph of Page 3, the output of Da_psi should be real and the ideal discriminator output is {0,1}.

In Algorithm 1, why the policy pi is related to Dc_phi.

Theorem 1 might not be right. A GAN matches two distributions. Besides, if E() is a linear function, delta could be a vector in the null space of the weight.

Many typos exist. For example, PGD-L_{inf} in Page 5.

In the last paragraph of Page 7, the representation sensitivity Sc is not defined.

---

> ### Author Response · Authors · 2020-11-20
> **Fixed the inconsistencies, and notation errors + improved clarity**
>
> We want to thank the reviewer for valuable comments and for pointing out several notation and clarity issues with the manuscript. We have made changes to the text and the figures to improve clarity and to address the concerns and previous errors. We hope that these changes would adequately address all of the reviewer’s concerns. We remain open to further discussing any possible remaining points.
> - **Structure and the oversized appendix**: indeed, the appendix includes many experimental results that we could not fit within the 8-page limit. These additional experiments were conducted to better demonstrate different aspects of the proposed method. Because of the 8-page limitation in the submission, we chose to only include the key results in the paper and to put the rest of the results in the appendix. However, all parts of the supplementary material are discussed in the main text and only the additional figures and tables are placed in the appendix. We believe that the extensive analyses included in the appendix should not be viewed as a downside of our paper but on the contrary as a positive sign of the thoroughness of our analyses. We are open to editing the structure of the manuscript and would appreciate it if the reviewer could provide more specific suggestions about the parts of the appendix that should be moved into the main text or vice versa.
> - **Revising figures and captions**: We interpret the reviewer’s request to “revise the figures and captions'' as a suggestion to improve the clarity of the figures and captions. We edited all the figures and captions to improve the clarity and readability. We are open to making more specific changes to the figures if the reviewer has any further suggestions.
> - **Claim of outperformance**: we agree with the reviewer that the previous statement might have been too strong. Hence, we edited this statement (pg2) to address the reviewer’s concern, replacing it by the following: “… show that our proposed approach performs similar or better  (often, significantly better) than most previous defense methods under most tested circumstances.". Given the results presented in Table-2, Table-3, Figure-2, and Figure-3 which show that AFD on many occasions significantly outperforms the baseline methods, we believe the updated statement accurately describes the results.
> - **Likelihood definition mismatch**: we thank the reviewer for pointing out this inconsistency in the definition of the likelihood function in the text. We changed the equation in Algorithm 1 to match that in the text.
> - **Inconsistent definition of \pi**: we thank the reviewer for pointing out this inconsistency. We fixed the \pi definition throughout the paper and included x and \epsilon as the two inputs $\pi(x, \epsilon)$.
> - **Formal definition of Da output**: The reviewer correctly pointed out this error in designating the output of the Da as defined in the paper. We corrected the definition by changing it to: “$\mathcal{S}(Da_{\psi})$ be a discriminator function $\mathcal{H}\rightarrow \{0,1\}$ that distinguishes between natural and perturbed representations and where $\mathcal{S}$ is the softplus function”.
> - **Relation between \pi and Dc**: The \pi function operates on the image using a specific class likelihood function which itself depends on the E and Dc functions. For this reason, we had included E and Dc functions in the definition. Following the reviewer’s comment, we noted that it would be more clear to define \pi simply as a function of x and epsilon $\pi(x, \epsilon)$. We changed the definition of \pi function in Algorithm 1 to be consistent with the rest of the text.
> - **Theorem1 might not be right**: The reviewer raises two concerns that we respond to each individually: 1) *GAN matches distributions*: indeed the adversarial learning procedure (i.e. GAN) matches two distributions of inputs. Proof of Theorem1 builds on top of this fact and shows that if the Dc classifier implements a Bayes optimal classifier then the claim of Theorem 1 is correct. 2) *the case where E is a linear function and delta could be a vector in the null space of the weight*: our understanding of this raised point is that if the delta vector indeed lies in the null space of the weight then it would be mapped onto zero by the E function which means that the embedding would remain unperturbed. Under this assumption delta would remain non-adversarial since it will fail to change the embedding by any amount. We hope that these explanations would address the reviewer’s concern and are still open to discuss further if any part of the concern is not addressed by our comment.
> - **Typos**: we proof-read the text again and specifically changed all L_{inf} instances to L_{\infty} for better consistency.
> - **Missing definition of sensitivity**: We thank the reviewer for pointing out the missing definition. We have now added the definition of sensitivity in the text on pg7.

---

### Official Review · AnonReviewer4 · 2020-10-25
**Review #4**

**Rating:** 6
**Confidence:** 5

**Review:**

Summary:

This paper proposes Adversarial Feature Desensitization (AFD) as a defense against adversarial examples. AFD employs a min-max adversarial learning framework where the classifier learns to encode features of both clean and adversarial images as the same distribution, thereby desensitizing adversarial features. With the aim of fooling a separate discriminator model into categorizing the classifier’s adversarial features as from clean images, the classifier is trained with the standard cross-entropy loss and adversarial loss terms. The authors showed through experiments on MNIST, CIFAR10 and CIFAR100 datasets that AFD mostly outperform previous defenses across different adversarial attacks under white- and black-box conditions.

Pros:
+Strong defense performance
+Novel idea

Cons:
-No discussion on the scalability of AFD defense or results on larger dataset such as imagenet

Recommendation:
The idea is interesting and is backed by strong empirical results. Generally, the paper is well-written and easy to follow. Given AFD employs both a GAN training component and adversarial example generation, I can imagine it to be computationally more expensive than most of the existing defenses. It would be more convincing to show experiments and discussions addressing AFD’s scalability. Apart from this point, there is no major flaw in the paper and I am inclined towards acceptance.


Other questions and comments:
For the black-box attack assessment, are the adversarial examples generated from models that are trained on their respective defense, initialized with different random seeds? It would be good to mention how exactly the black-box attacks are conducted.

Some prior defenses have been shown to be prone to transfer black-box attacks, e.g. adversarial examples from a TRADES model, transferred to the AFD defense. It would be more convincing that AFD is not relying on obfuscated gradients with results on this form of black-box attacks.

How much is AFD’s performance dependent on the strength of adversarial examples using during training? How many iterations were used for the L-inf attack policy used to perturb the training inputs?

I believe the sentence in the Introduction is untrue: “adversarial learning has not yet been successfully applied to the problem of adversarial robustness.” Several studies using adversarial learning for adversarial robustness exist:

“Improved Network Robustness with Adversary Critic” NeurIPS 2018

"A Direct Approach to Robust Deep Learning Using Adversarial Networks" ICLR2019

“What it Thinks is Important is Important: Robustness Transfers through Input Gradients” CVPR 2020”

“GanDef: A GAN based Adversarial Training Defense for Neural Network Classifier” arXiv:1903.02585

--Update after rebuttal--

The reviewer thanks the authors for addressing the key questions and concerns and have updated the confidence score accordingly.

---

> ### Author Response · Authors · 2020-11-20
> **Analysis of scalability and computational cost + transfer black-box attacks**
>
> We want to thank the reviewer for the suggestions and comments. We especially found the suggested analysis on the relation between robust performance and the attack strength very insightful which led to an interesting observation. We found that the our proposed approach was able to retain most of its robust performance against PGD-based attacks using substantially weaker attacks (PGD-L_inf with 5 iterations and eps=4/255 on CIFAR10) while showing significantly higher robust accuracy against other attacks like C&W and DeepFool. We think that using weaker attacks during training leads to less overfitting to the specific form of attack used during training. This also has important implications on the scalability of the proposed method since most of the computational cost of training is associated with generating the adversarial examples with default number of iterations.
> - **Scalability**: The reviewer is correct in commenting on the computational cost of the proposed method. As mentioned by the reviewer, AFD uses two additional loss terms from adversarial training and in principle it involves 3 back-propagation steps for each batch. Despite this, we found that most of the computational cost associated with training is spent on generating the adversarial examples which involves N (number of iterations) back-propagation steps per batch. On that point, the new experiments that we conducted (see the below point on performance vs attack strength) show that AFD is capable of retaining most of the previous robustness when using substantially weaker attacks during training. This suggests that the computational cost associated with AFD could potentially be drastically  reduced by using weaker attacks with significantly less number of iterations. We have added these new findings as a discussion point in the paper.
> - **Imagenet results**: We agree with the reviewer on the importance of scalability of the proposed method to larger datasets. Our choice of datasets which we used to evaluate our method was based on the feasibility of training as well as ease of comparison with alternative methods. To accommodate the reviewer’s comment, we are currently running our method on the tiny-imagenet dataset and hope that we can finish this experiment in time to report the results before the end of the discussion period.
> - **Black-box experiment details**: The black-box adversarial images were produced by running the attack on a naturally trained ResNet18 model (same architecture as in AFD) and evaluating the resulting images on all baseline models. We added this information to the text.
> - **Transfer black-box attacks**: Following the reviewer’s suggestion, we also tested the transfer black-box attacks from adversarially trained (AT) models on all three datasets. The results showed that the AFD model is still more robust than the baseline models in response to these attacks. We added these results to the Table-A3 in the appendix.
> - **Performance vs. attack strength**: Following the reviewer’s suggestion we tested the effect of attack strength on robust performance in AFD by parametrically changing the number of attack iterations (5, 10) and the maximum perturbation distance (epsilon=4/255, 8/255) on CIFAR10 dataset. We found that using particular weaker attacks during training (4/255, it=5), AFD could retain most of its robustness against a large set of attacks while improving its robustness against C&W and DeepFool attacks compared to before. We added these results as a supplementary figure in the appendix.
> - **Inaccurate claim**: We thank the reviewer for bringing this issue to our attention. We agree with the reviewer that the claim in that statement is inaccurate when considering several other papers that have used adversarial learning in the same context. Our goal was to communicate that this particular form of adversarial robust representation learning had not been tried before. We clarified this point by editing the introduction text and adding the missing references.

---

### Official Review · AnonReviewer2 · 2020-10-29
**This paper proposes adversarial feature desensitization to generate robust representation**

**Rating:** 5
**Confidence:** 5

**Review:**

This paper proposes to leverage an additional adversarial discriminator to distinguish between the clean and perturbed inputs from the representation level.

The empirical evaluation shows promising results in terms of generalizing to unforeseen attacks.
However, the baselines TRADES is evaluated on ImageNet which is a more challenging task and it would be good to evaluate the performance on large scale dataset for the proposed method as well.
It would be important to compare with related work on improving robustness by learning robust representation [1][2].
In addition, from figure 4, the sensitivity comparison is actually not very clear and it would be good to compute the significance level to show how sensitive the AFD learned representation with quantitative results.

From the methodology perspective, it would be good to analyze the difference between the distributions of benign and adversary representation. Several previous work shows that with only activation patterns or logits it is not enough to distinguish the benign and adversarial instances. It would be important to evaluate and confirm that the learned representation of the benign and adversarial instances can indeed be learned and separated by a trained discriminator, which may lead to further interesting analysis and findings.


[1] Liao, Fangzhou, et al. "Defense against adversarial attacks using high-level representation guided denoiser." Proceedings of the IEEE Conference on Computer Vision and Pattern Recognition. 2018.
[2] Samangouei, Pouya, Maya Kabkab, and Rama Chellappa. "Defense-gan: Protecting classifiers against adversarial attacks using generative models." ICLR.

---

> ### Author Response · Authors · 2020-11-20
> **Analysis of the adversarial discrimination and more experiments on tiny-imagenet**
>
> We want to thank the reviewer for its valuable comments and for suggesting more analyses that we believe have strengthened our paper. Notably, we found the suggested analysis of the performance of the adversarial discriminator very insightful. We found that for all three datasets the discriminator was indeed able to discriminate between the clean and perturbed embeddings. However, this ability was asymmetrical between the images from the two distributions and this ability was diminished with more training. We have made several changes to the manuscript to address the reviewer’s concerns and hope that these changes have addressed the important concerns raised by this reviewer.
> - **ImageNet results**: We agree with the reviewer on the importance of scalability of the proposed methods. Because of the high computational demands of our proposed method we evaluated its performance on several datasets that are commonly used in the adversarial robustness literature (MNIST, CIFAR10 and CIFAR100). To accommodate the reviewer’s request, we are currently running our method on the tiny-imagenet dataset and hope that the results will be ready before the end of the discussion period. We will update our response once we have those additional results.
> - **Comparison with more related work**: We thank the reviewer for suggesting the two additional publications. We added the missing references to the paper and added Defense-GAN result on MNIST to Table2 in the updated manuscript.
> - **Significance of differences in Fig4**: We thank the reviewer for this suggestion. We tested the significance of differences between the TRADES and AFD methods. Because of the unequal variances between AFD and AT, we used the Welch’s t-test to test the significance of difference. We found that for most comparisons, the differences between the means were significant. We indicated this information on the subplots in Fig4.
> - **Da performance**: We thank the reviewer for this suggestion. Following the reviewer’s suggestion, we evaluated the accuracy of Da classifier on several checkpoints during the training for all datasets. The results show that during the training, the Da classifier can successfully distinguish between the clean and adversarial images from their embeddings. Although the classification accuracy is asymmetric for adversarial and clean images which is potentially due to the asymmetric training approach we have chosen. Unlike the previous work mentioned by the reviewer, in our method the adversarial discriminator also receives the class labels as inputs and therefore learns to distinguish between the clean and adversarial inputs conditioned on those labels. We added plots of adversarial decoder accuracy as a supplementary figure (Fig-A5).

---

### Official Review · AnonReviewer3 · 2020-11-01
**Not enough novelty with potential problems in the evaluation.**

**Rating:** 4
**Confidence:** 5

**Review:**

The paper proposes an adversarial defense method that builds a robust classifier by using a min-max optimization in the feature extractor.

Pros:
1. The proposed idea is interesting and reasonable.
2. The experiment results look good.

Cons:
1. The proposed idea is not novelty enough. There is a similar work [1] talking about the feature scattering to make the model robustness.  The only difference is the paper further uses a discriminator which involves a different network in the training process.
2. The results might not be reliable. while [1] uses a similar idea, it later shows a significant performance drop by using some later proposed adversarial attack method like Autoattack[2]. Specifically, it reduces the performance to from claimed 60.6to 36.64. I suggest the author should try more blackbox based attack or the ensemble attack in Autoattack.
3. The paper quality needs to be improved. For example, Figures 3  doesn't have x-axis label. I personally find it very confusing on the different x scale used in the figure.
4. The methods mentioned in non-obfuscated gradients are not sufficient for me to believe there is no such case. (ii) (iii) is quite unrelated to the obfuscated gradients problem. Also, the boundary attack in the appendix looks very confusing. I can't understand what the number represents in the table. By the way, the boundary attack is not a good way to attack the adversarial defended method. Attacks like B&B or FAB are more suitable.





[1] Zhang, Haichao, and Jianyu Wang. "Defense against adversarial attacks using feature scattering-based adversarial training." Advances in Neural Information Processing Systems. 2019.

[2] https://github.com/fra31/auto-attack

---

> ### Author Response · Authors · 2020-11-20
> **Important differences between AFD and Feature-Scattering method**
>
> We want to thank the reviewer for its insightful comments and proposed additional evaluations. We provided a detailed comparison of the differences between the feature-scattering method and our proposed method to clarify the novel contributions of our method. We also tested our proposed method against additional attacks as suggested by the reviewer and reported the results in the manuscript. We believe that these new evaluations have improved the strength and transparency of our paper and have further highlighted the strengths and weaknesses of our method.
> - **Novelty compared to feature-scattering method**: Although the paper cited by the reviewer (feature-scattering) has some similarities with our proposed method at the conceptual level, we want to note that there are important and critical differences between that method and our proposed one. Specially, 1) the feature scattering method considers representational differences as a constraint during the adversarial example generation and it uses adversarial training to learn to defend against that class of examples. In contrast, our proposed method defends against distributional differences between clean and perturbed image embeddings directly and uses standard PGD-L_{inf} adversarial attack during training. 2) the feature scattering method considers a norm-based regularizer to reduce the representation differences during the attack - which is completely different from our proposed method that depends on an adversarial learning procedure to discriminate between distributions of clean and perturbed embeddings (in contrast to sample-wise differences). On that point, we also had compared the defense method based on standard L2 distance with our proposed adversarial learning procedure in Fig.A3 which confirmed that the adversarial learning approach could be far more superior compared to the norm-based distance minimization.
> - **Evaluation on AutoAttack**: We thank the reviewer for suggesting to additionally evaluate our method on AutoAttack. We tested the AFD-trained network on the AutoAttack ensemble attack and the results were similar to those found for C&W and DeepFool attacks. The AFD-trained model performed better than the baseline models on the MNIST dataset but worse on the CIFAR10 and CIFAR100 datasets. We want to emphasize again that our proposed approach does not guarantee generalization to all attack types and the degree of generalization would depend on the differences in the distribution of embeddings seen during training and test phases. We added these results to Table-A3.
> - **Fig3 axis labels**: We want to thank the reviewer for bringing this point to our attention. Because this figure includes 8 subplots with the same axis labels we omitted the axis labels on all subplots except one (the lower left one). We understand that this might be confusing to some readers and for that reason we edited the figure and repeated the axis labels on different locations to help with the readability. In addition, regarding the point about different ranges on the x-axes — we wanted to note that the epsilon range was selected so that all attacks would substantially reduce the accuracy for most tested methods.
> - **Insufficiency of tests for showing non-obfuscated gradients**. There are several points raised by the reviewer in this topic that we respond to individually. 1) the reviewer mentions that points (ii) (iterative vs single-step attack success) and (iii) (black-box vs white-box attack success) are unrelated to the obfuscated gradients problem. However both of these points along with several others are listed in [1] as recommended analyses for correct evaluation of adversarially robust models. To our knowledge the analyses in [1] are widely accepted in the community and we also attempted to follow most of these guidelines in our work. 2) *Confusing results on the Boundary Attack*: we apologize if the reported numbers for the Boundary Attack were confusing to the reviewer in any way. As mentioned in the table caption, the reported numbers show the robust accuracy of each model on random samples selected from each dataset’s test-set. We further edited the table caption to improve clarity. 3) *More black-box evaluations are needed*: We thank the reviewer for suggesting to additionally evaluate our method against other black-box attacks. Following the reviewer’s suggestion, we additionally evaluated our model on the B&B attack and compared our results with other baseline models. The results were mixed across different datasets with AFD performing better than all baselines on the MNIST dataset, better than AT but worse than TRADES on CIFAR10, and worse than both baselines on CIFAR100 dataset. We added these results to Table-A3 in the appendix.
>
> [1] Carlini, N., Athalye, A., Papernot, N., Brendel, W., Rauber, J., Tsipras, D., ... & Kurakin, A. (2019). On evaluating adversarial robustness. arXiv preprint arXiv:1902.06705.

---

### Decision · Program_Chairs · 2021-01-07
**Final Decision**

**Decision:**

Reject

**Comment:**

This paper proposes Adversarial Feature Desensitization (AFD) as a defense against adversarial examples. Specifically, following the spirit of GAN and Adversarial Domain Adaptation, an adversarial discriminator is introduced to distinguish clean and perturbed inputs at the representational level.

This paper receives 3 reject and 1 accept recommendations. On one hand, though the proposed method shares some similarity with the Feature Scattering method at a high level, most of the reviewers still find the proposed method is interesting. The AC also agrees that the paper's organization and typos does not warrant a rejection.

On the other hand, the reviewers have also raised a few concerns. (i) A more careful discussion on the scalability of the proposed method is needed. (ii) Experiments are mostly focused on small datasets, while results on ImageNet is lacking, which makes the paper less convincing. The authors claim that they are trying to at least run Tiny-ImageNet experiments; however, this set of results are not provided by the end. (iii) A more detailed analysis and visualization on the learned difference between the distributions of benign and adversary representation is needed, since a discriminator is learned here.

The rebuttal unfortunately did not fully address the reviewers' main concerns. On balance, the AC regrets that the paper cannot be recommended for acceptance at this time. The authors are encouraged to consider the reviewers' comments when revising the paper for submission elsewhere.